# Exercise-based cardiac rehabilitation and atherosclerotic plaque regression in ASCVD: Is exercise really a game-changer? A scoping review of controlled trials

Dwita Rian Desandri [1,2]*, Averina Octaxena Aslani [2], Faqrizal Ria Qhabibi [2], Dimitra Nikoletou [3]

1 Department of Cardiology and Vascular Medicine, Faculty of Medicine, Universitas Indonesia, Jakarta, Indonesia, 2 Cardiovascular Prevention and Rehabilitation Division, National Cardiovascular Center Harapan Kita Hospital, Jakarta, Indonesia, 3 School of Health and Medical Sciences, City St George's, University of London, London, United Kingdom

* dwita.desandri@pjnhk.go.id

## Abstract

Exercise-based cardiac rehabilitation (CR) has emerged as a promising strategy for promoting atherosclerotic plaque regression in patients with atherosclerotic cardiovascular disease (ASCVD). Although lipid-lowering drugs are firmly established for plaque regression, the impact of exercise-based CR on plaque regression remains underexplored, prompting synthesis of existing evidence. This review aims to characterize studies on exercise-based CR in ASCVD, summarize CR modalities, identify confounding factors, and evaluate evidence of plaque regression from imaging and clinical outcomes. A thorough literature search was conducted in May-June 2025 across PubMed, ScienceDirect, and EBSCO. Included studies were controlled trials on CR for ASCVD adults assessing plaque changes via imaging. Data extraction followed PRISMA-ScR guidelines focusing on study design, populations, interventions, and outcomes. Five trials encompassing 217 participants demonstrated that diverse CR programmes—high-intensity interval training, aerobic exercise and resistance training—combined with standard medical therapy significantly reduced plaque volume parameters, as measured by IVUS and carotid ultrasound. CR further improved key mechanistic markers, including inflammatory cytokines and endothelial function, underpinning plaque stabilization and regression. Exercise-based CR robustly promotes plaque regression and mitigates cardiovascular risk in ASCVD, complementing pharmacological approaches. These findings affirm CR's indispensable role in secondary prevention, though confounding from concurrent lipid-lowering therapies warrants consideration in future randomized trials.

which permits unrestricted use, distribution, and reproduction in any medium, provided the original author and source are credited.

**Data availability statement:** All relevant data are within the manuscript.

**Funding:** The author(s) received no specific funding for this work.

**Competing interests:** The authors have declared that no competing interests exist.

## Introduction

Prolonged exposure to pro-inflammatory atherogenic stimuli associated with conventional and non-conventional cardiovascular risk factors leads to endothelial dysfunction, facilitating the migration of low-density lipoprotein-C (LDL-C) particles across the endothelium and their subsequent retention and oxidation within arterial vessels. This event initiates atherosclerosis formation. The atherosclerosis does not grow overtime, instead it alternates episodically between phases of rapid growth, dormancy, and rupture or erosion.

Reversing the natural progression of atherosclerosis would be feasible by achieving plaque stabilization and regression. To attain plaque regression, high-risk plaque features must be eliminated by not only reducing the total plaque volume, but also modifying plaque components and changing the morphology from noncalcified to calcified plaque features [1, 2].

Various strategies have been proposed for plaque regression, with statins being the most extensively researched [3–5]. Recent studies have also mentioned other lipid agents like ezetimibe and proprotein convertase subtilisin/kexin type-9 (PCSK-9) inhibitors [6–8], as well as agents with less favourable effects like colchicine and eicosapentaenoic acid [9–11].

While exercise is correlated with a decreased coronary artery disease (CAD) risk, the precise mechanisms by which exercise training affects plaques remain uncertain. It is thought to be the improvements in multiple metabolic derangements and cardiovascular risk profiles including body weight, lipid profile, blood glucose, and exercise capacity [12–14]. Exercise also could improve vasodilation through enhancing intrinsic nitric oxide (NO) synthase and reducing reactive oxygen species (ROS) and inflammation in CAD [15].

Exercise training, either alone or in the context of exercise-based cardiac rehabilitation (CR), is an established secondary prevention in atherosclerotic management [16]. A study from Madsen et al. revealed that aerobic exercise, whether it is interval or continuous training, and medical therapy for at least 12 weeks resulted in moderate regression of necrotic core and plaque burden in intravascular ultrasound (IVUS)-defined coronary lesions [17]. Recent studies including several randomized controlled trials (RCTs) have been published since then. Therefore, this paper is subjected to critically synthesize evidence from recent studies on exercise-based CR and its impact on atherosclerotic plaque regression in patients with atherosclerotic cardiovascular disease (ASCVD).

## Methods

### Search strategy

The search was conducted from May 23 to June 1, 2025, utilizing three electronic databases: PubMed, ScienceDirect, and EBSCO. To ensure comprehensive coverage, a manual review of retrieved articles was also performed to find additional studies that may have been overlooked in the database search. Search terms combined medical subject headings (MeSH) and keywords without abbreviations to ensure clarity, using Boolean operators such as: "cardiac rehabilitation" AND "plaque regression"

OR "atherosclerotic regression"; "cardiac rehabilitation" AND ("plaque regression" OR "atherosclerotic regression"). Three independent reviewers (D.R.D., A.O.A., F. R. Q.) independently screened titles and abstracts, then assessed full texts for eligibility. A fourth reviewer resolved any discrepancies.

### Eligibility criteria

Studies included were peer-reviewed articles that met the review's inclusion criteria. The studies had to focus on examining CR programs for patients with atherosclerotic cardiovascular disease (ASCVD) aged 18 and above, using IVUS or any imaging modality, whether invasive or non-invasive, to assess plaque regression using PAV and TAV parameters. The analysis is limited to English-published articles to ensure accessibility and inclusion of widely accessible research. Eligible study designs comprised randomized controlled trials (RCTs) or non-RCTs. Studies were excluded if they lacked detailed descriptions of CR modalities, frequency, demographics, or if they were case-based research or non-trial designs. If an article with potentially overlapping study populations is identified, the relevant author will be contacted for clarification.

### Data extraction

This scoping review was conducted following the Preferred Reporting Items for Systematic Reviews and Meta-Analyses Extension for Scoping Reviews (PRISMA-ScR) framework [18]. These scoping reviews were not registered in PROS-PERO or any other database, as they are purely exploratory in nature rather than systematic reviews with pre-established protocols. Extracted data included the first author; year of publication; research design; country or place of research; population characteristics that include population size, gender proportion, and average age of the population; modalities of CR programs in the intervention group; treatment of the control group; modality of plaque/atheroma assessment; and study outcomes. Data were extracted independently by three reviewers (D.R.D., A.O.A., F.R.Q.) and compiled into a results table in Microsoft Excel for Mac. Given the review's aim to map available evidence, the absence of a formal risk of bias assessment or quality appraisal tool was deemed appropriate. Grey literature, such as unpublished studies and conference abstracts, was searched, but no relevant records were identified according to the applied search terms.

## Results

### Literature search

The PRISMA flowchart is presented (Fig 1). We conducted a comprehensive literature review and selected 425 results from three databases: PubMed (n = 44), ScienceDirect (n = 231), and EBSCO (n = 150). Early screening removed duplicate records (n = 23), removed papers with inappropriate abstracts and titles (n = 67), non-English publication (n = 1), and inaccessible full-text articles (n = 172). Therefore, we have reduced the selection to 162 research studies that will be subjected to eligibility assessment. Out of 162 studies, 15 were removed owing to unsuitable population characteristics, 98 due to incorrect evaluation of parameters and outcomes, 27 due to unsuitable study design, and 17 papers were excluded due to wrong intervention. This strategy yielded five studies, and the recruited studies consisted of 217 individuals who participated in three randomized controlled trials (RCTs) and two non-randomized controlled trials (non-RCTs).

### Characteristics of population and included studies

Five studies involved 217 participants from four countries and three continents. The population consisted of individuals with atherosclerotic cardiovascular disease (ASCVD), including chronic coronary syndrome (CCS)/stable angina pectoris, patients with spectrum acute coronary syndrome (ACS) who underwent primary percutaneous coronary intervention (PPCI), and patients with a history of transient ischaemic attack (TIA) or mild non-disabling stroke. Secondary prevention medications, such as lipid-lowering therapies (LLTs), anti-anginal, anticoagulant, and risk factor-controlling medications like hypertension and diabetes, were administered according to treatment standards. In the intervention group, patients

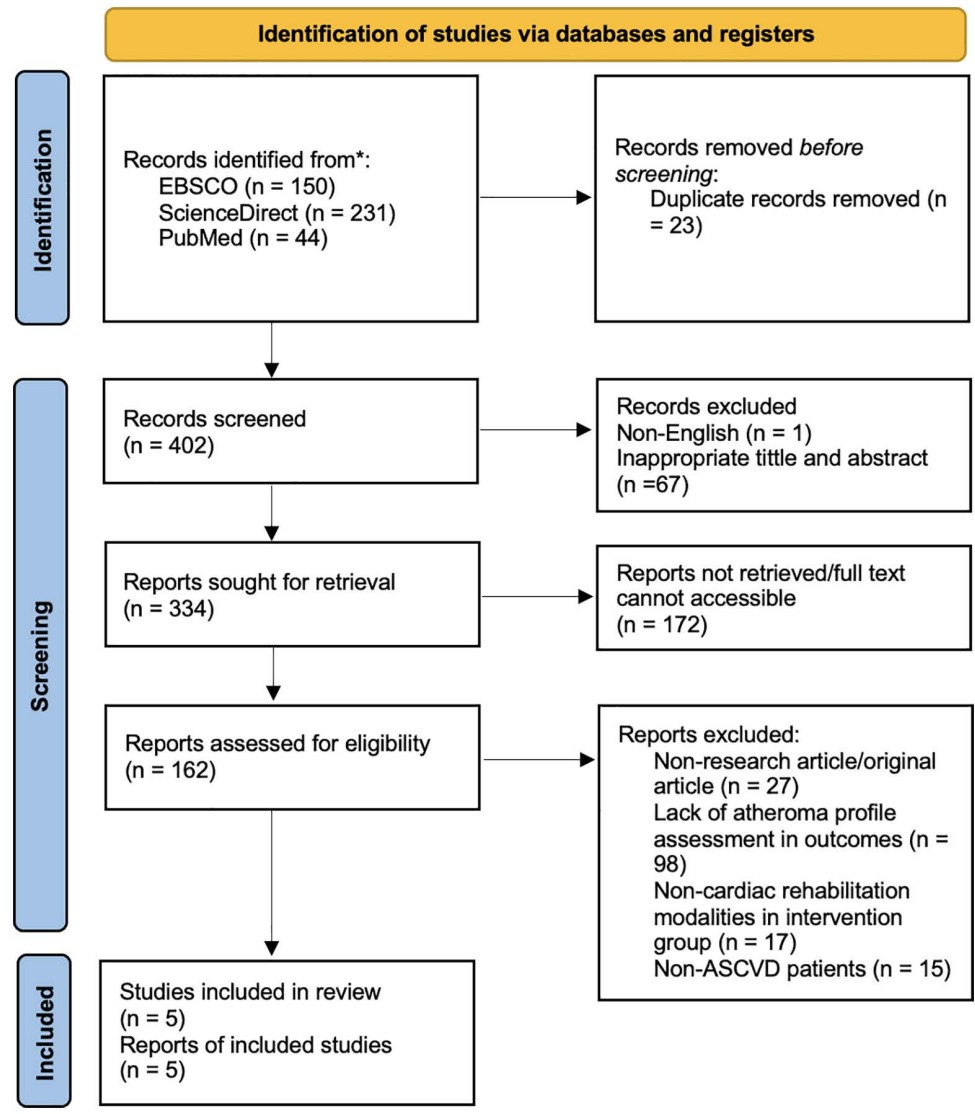

**Fig 1. Literature search flow chart by PRISMA Guideline 2020.**

received standard medications and a CR programme that varied across studies. Detailed study characteristics are presented in Table 1.

## Cardiac rehabilitation programmes

In the five *included studies,* there were variations in CR programs carried out in the intervention group. In the first study by Vesterbekkmo E.K., et al., the intervention group was given a high-intensity interval training (HIIT) program with a frequency of 2 times a week [19]. Each session begins with a 10-minute warm-up at moderate intensity (60–70% of peak heart rate (HR)), followed by 4x4-minute intervals at 85–95% of *peak HR* with 3 minutes of active recovery at moderate intensity between intervals, ending with a 5-minute cool-down period.

Moreover, a study by Kurose, S., et al. integrated aerobic exercise (3 times a week, each 30 minutes with a bicycle or treadmill at the anaerobic threshold) and resistance training (push-ups, sit-ups, and squats modified with body weight; 3

**Table 1. Extraction data and characteristics of included studies.**

| Author | Year | Study Design | Place/Country | Population Characteristics | Interventions | Control | Assessment | Outcomes |
|---|---|---|---|---|---|---|---|---|
| Vesterbek-kmo E.K., et al. [19] | 2022 | open-label RCT | Norway | Stable angina patients undergoing PCI, N total: 59 patients (M: 54, F: 5); mean age of intervention group: 57.3±6.8 years, mean age of controlled group: 58.7±7.4 years. | HIIT at 85–95% of peak HR with 2 sessions weekly | Followed contemporary preventive guidelines but did not receive any supervision nor wearable device for physical activity tracking | IVUS | PAV in intervention group decreased from 49.5±9.4% to 48.3±8.4%, while in controlled group PAV remained constant at 48.6±6.1% and 48.9±6.0% for baseline and follow-up. TAV decreased (mm$^3$) in intervention group from 162.7±60.5 to 154.9±57.2, while in controlled group, TAV increased from 179.4±61.4 to 182.1±63.3. Minimal lumen area/MLA (mm$^2$) in intervention group remain constant between baseline and follow-up, however in controlled group there was a narrowing MLA from 47±1.3 to 4.4±1.4. |
| Kurose, S., et al. [20] | 2015 | Non-RCTs | Japan | ACS patients undergoing emergency PCI, N total: 41 patients with mean age of intervention group was 63.1±9.1 years and control group was 61.3±8.4 years. | Exercise-based CR | Non-CR groups | IVUS | Decreased PAV (mm$^2$) 10.1±4.6 to 5.7±8.3 in intervention group, while in controlled group, PAV increased from 5.3±4.5 to 6.4±6. MLD (mm) in both group remain constant between baseline and follow-up, nevertheless in the intervention group MLD slightly larger than control group. |
| Yokoyama M.N., et al. [21] | 2019 | open-label RCT | Japan | ACS patients who admitted between February 2013 and January 2016, N total: 32 (M: 30, F:2); mean age of intervention group was 58±10.2 years, controlled group was 59.8±9.8 years. | Intensive CR (late phase II CR, ≥ twice per week and step count ≥9,000/day). | Standard CR group (late phase II CR ≥once/2 weeks and step count ≥6,000/day). | IVUS and Integrated Backscatter IVUS (IB-IVUS). | Decreased PAV (mm$^2$) was 87.2±59 to 78.4±54.5 (−8.9±14.2) in the intensive CR group, the standard CR group also presented PAV decrease 62±43.8 to 58.5±41.2, however, their reduction was smaller than intensive group (only −4.5±5.5). |
| Yokoyama M.N., et al. [22] | 2015 | Non-RCTs | Japan | ACS patients who admitted between December 2009 and August 2012, N total 46 (M: 44, F: 2); mean ages of intervention group was 61.7±10.2 years, control group was 58±9.8 years. | Phase II CR after discharge. | Non-phase II CR | 3D-IVUS | PAV (mm$^2$) decreased from 112±59.5 to 107.9±62 (difference=−4.5±10.7%) in the intervention group, whereas control group also decreased from 103.8±84.8 to 98.6±79.5, however their reduction was smaller than intervention group (only −3±13.3%). |
| Linden-maier T.J., et al. [23] | 2013 | open-label RCT | United Kingdom (London) and Canada (Ottawa) | History of TIA or mild non-disabling stroke, N total 39 (M: 24, F: 15); mean ages of intervention group was 64±10 years, control group was 66±11 years. | 6 months comprehensive CR (CCR) | Non-CCR | Carotid ultrasound (US) | Total plaque volume (mm$^3$) reduction of carotid atherosclerosis in the intervention group from 55±60–53±57, while the control group exhibit an increase from 137±155–138±116. |

sets of 10–15 repetitions on the Borg index of 11–13, respectively, 3 times a week) in their CR programme [20]. In another RCT study by Yokoyama M.N., et al. from 2013–2016, the intervention group was given intensive CR, namely late phase II CR participation, ≥ 2 times per week and the number of steps ≥ 9,000/day [21]. Meanwhile, the control group was given a standard CR of late phase II CR participation, ≥ once/2 weeks and the number of steps ≥ 6,000/day.

A different study (non-RCT) from the same researcher and conducted in 2009–2012 in ACS patients, the intervention group was given a CR program consisting of warm-ups-stretches, 60 minutes of aerobic exercise, resistance training, and cool-down, and was scheduled once or twice a week for 6 months [22]. From the last study by Lindenmaier, et al., comprehensive CR was carried out in patients with a history of TIA or mild non-disabling stroke (MNDS) [23]. It consists of 50 sessions carried out 2x/week, with additional training in the form of home-based training at least twice a week. Exercise follows a progressive individual prescription, an HR reserve of 40% to 70%, or Borg Index of 11–14 [24].

## Confounding factors

Cardiac rehabilitation programmes do not solely influence the regression of atherosclerosis plaques. Other factors that could affect plaque development may also serve as confounding variables in the main idea of this paper, including the prescription of statins or other LLTs; individual risk factors such as hypertension, diabetes mellitus, dyslipidaemia, and smoking; and lastly hereditary factors like familial hypercholesterolaemia.

To identify whether plaque regression is significantly influenced by the CR programme, the identification of confounding factors for each included study should be analysed in both the intervention group and the control group. Vesterbekkmo et al. showed that both groups received prescriptions for statins and even had a balanced proportion of individuals who received additional ezetimibe prescriptions, namely 2 individuals (7%) in the intervention group and 3 individuals (10%) in the control group [19]. When looking at the proportion of risk factors in both groups, a balanced distribution was observed, including hypertension (55% vs. 53%), previous smoking (55% vs. 53%), DM (7% vs. 17%), and congestive heart failure (4% vs. 7%). When comparing the lipid profiles before and after follow-up in both groups, the data indicate that there were no statistically significant changes in LDL-C, HDL-C, triglycerides, and cholesterol levels.

Study conducted by Kurose S. et al. also showed almost similar profile related to the proportion of statin prescriptions, namely 14/21 subjects and 16/20 subjects in the intervention and control groups, respectively [20]. LDL cholesterol, LDL/HDL ratio, and Hs-CRP levels reductions were all observed in both groups. Nevertheless, the IVUS profile showed that plaque regression only happened in the intervention group, whereas plaque growth occurred in the control group that did not receive the CR program. The level of medication adherence of each subject is crucial to exclude the presence of confounding factors, but these data are not listed in all included studies, constraining the authors to rely on the existing data; namely the baseline data and measured parameters comparing pre- and follow-up lipid profiles that are closely related to the efficacy of statins, one of the strongest confounding factors. All studies did not report colchicine and EPA use.

## Plaque regression

Plaque regression was observed in all four studies that used IVUS [19–22] and one study that used carotid ultrasound [23]. Vesterbekmo et al.'s study revealed a reduction in mean PAV from 49.5 ± 9.4% to 48.3 ± 8.4% in the intervention group, which incorporated HIIT alongside standard treatment, demonstrating a statistically significant reduction (p 0.017) [19]. In contrast, the control group experienced an increase in average PAV from 48.6 ± 6.1% to 48.9 ± 6%, resulting in a marginal rise of approximately 0.2% (p = 0.616). Moreover, the intervention group showed a reduction in TAV by 9 mm³, whereas the control group exhibited an increase in TAV by 3 mm³ (p = 0.002 and p = 0.616, respectively).

In a subsequent study by Kurose et al., the intervention group that participated in the CR programme demonstrated a significant decrease in PAV of 4.3 ± 9.8 mm³ compared to the control group that did not engage in the CR programme [20]. Furthermore, the intervention group experienced a significant reduction in plaque area by 1.6 ± 3.4 mm², while the control group showed an increase (p < 0.05 and p < 0.01, respectively).

Another study by Nishitani-Yokohama M. et al. compared the intensive CR group with the standard CR group plus standard treatment [21]. The findings revealed that both groups exhibited PAV reduction; however, the intensive CR group experienced a significantly larger decrease, with a reduction of $8.9 \pm 14.2$ mm³, while the standard CR group only decreased by $4.5 \pm 5.5$ mm³. All test results were statistically significant with $p < 0.01$.

The final IVUS study by Nishitani-Yokohama M. et al. in 2015 compared the phase II CR group with a control group [22]. Both groups showed a reduction in PAV, although the intervention group experienced a greater decline ($-4.5 \pm 10.7$ mm³) compared to the control group ($-3 \pm 13.3$ mm³). Lastly, the study utilising carotid ultrasound indicated that the comprehensive CR group experienced a reduction in total plaque volume (TPV) by $2 \pm 14$ mm³, while the control group experienced an increase in TPV by $1 \pm 18$ mm³ [23].

## Discussion

Plaque atheroma regression is defined as a change in plaque volume burden and composition [2, 25] Numerous studies have used both PAV and TAV to assess plaque burden since it appears to be the logical definition of plaque regression [25, 26] Plaque regression emphasizes aggressive and long-term strategies: removing lipids and necrotic core materials, restoring normal endothelial function, and reducing inflammation beyond just stabilising plaques [2]. Chacon et al. explained about the potential mechanisms of exercise benefit in atherosclerosis: cardiac preconditioning by upgrading ROS scavenging, increased collateralization by expanding arteriogenesis, and plaque regression [27].

Studies have assessed the effects of CR on inflammatory markers like fibrinogen, TNF-α, IL-6, and CRP. A meta-analysis by Sadeghi M., et al., showed CR significantly lowered CRP levels and reduced inflammation in the short and long term [28]. CR decreases inflammation by diminishing macrophage activity, foam cell formation, platelet reactivity, and fibrous cap thickening. Some studies also demonstrated that exercise mobilizes T cells, natural killer cells, releases anti-inflammatory IL-10, and reduces visceral adipose tissue [29]. Taken together, CR could reduce ASCVD recurrence through positive effects on inflammatory conditions [30, 31] Kurose S. et al. confirmed the inflammatory pathway relation in plaque regression by CR, showing a significant decrease in hs-CRP levels from $0.427 \pm 0.519$ mg/dL to $0.063 \pm 0.060$ mg/dL after 6 months ($p < 0.01$). The intervention group also had a reduction in PAV ($4.3 \pm 9.8$ mm³) and plaque area ($1.6 \pm 3.4$ mm²) compared to the control group ($p < 0.05$ and $p < 0.01$, respectively) [20]. Vesterbekkmo et al. observed PAV and TAV reductions after 6 months of HIIT are comparable to statin trials [19].

Not only the inflammatory pathway, CR also has the ability to suppress endothelial dysfunction that has been proven as the initial step of plaque formation. Programmed physical exercise in CR increases bioavailability of nitric oxide and circulating endothelial progenitor cells, as well as decreases oxidative stress, eventually preventing endothelial dysfunction [32].

The mechanisms of removal of lipid and necrotic core consist of enhanced cholesterol efflux due to improved reverse cholesterol transport, foam cell removal, and macrophage shift towards 'reparative' M2 phenotype (promotes osteoblast differentiation and smooth muscle cell maturation, changing the plaque into a more stabilised and calcified form) [25]. Exercise in the long term has been shown to promote M1 to M2 macrophage switch [33] and reduce necrotic core [17]. Nevertheless, none of the studies assessed necrotic core volume.

Taken together, exercise-based CR produces all three effects that stimulates plaque regression. Several journals have discussed LLTs and their effect on plaque regression thoroughly [5,25,34,35]. However, exercise-based CR could act synergistically with the LLTs in plaque regression and should be considered as important as LLTs to stimulate plaque regression and eventually reduce CV risk.

There are several limitations in this review. Firstly, the current analysis was unable to independently distinguish the specific role of concomitant LLTs in the observed regression of atherosclerotic plaque, a factor that remains a potential confounder in evaluating the primary intervention. Lack of information on drug adherence prevents us from determining whether plaque regression is directly attributable to CR. Second, the relatively short duration of follow-up across the

included studies precluded the assessment of clinical endpoints and long-term cardiovascular outcomes. Third, significant heterogeneity in the CR regimens utilized across the literature makes it challenging to identify the most effective specific CR intervention. Lastly, although grey literature was considered, no relevant records were identified.

Based on the limitations identified in this study, future research should prioritize several key areas to bridge the current gaps in cardiovascular rehabilitation and plaque management, including (1) dissecting therapeutic synergy to accurately isolate the regression of atherosclerotic plaque. Future trials must be designed to distinguish the independent and effects of LLTs versus lifestyle-based CR intervention. (2) Long-term outcome data, there is a critical need for studies with extended follow-up duration in the aim to obtained hard clinical endpoints, including MACE, cardiovascular mortality, and all-cause hospitalization. (3) Future evidence synthesis would benefit from a higher volume of double-blind and multicenter RCTs, providing a clearer picture of the intervention's true efficacy. (4) Leveraging high-resolution intravascular imaging in future cohorts could provide deeper insights into plaque morphology and stability beyond simple volume regression.

## Conclusion

Exercise-based CR, regardless of its type and modality, influences atheroma plaque regression based on the plaque volume parameters reduction, although the five included studies could not definitively rule out the influence of lipid-lowering therapies and risk factors in the population due to limited data in the included studies.

## Supporting information

**S1 Table. PRISMA Checklist.**
(DOCX)

## Author contributions

**Conceptualization:** Dwita Rian Desandri.

**Data curation:** Dwita Rian Desandri, Averina Octaxena Aslani, Faqrizal Ria Qhabibi.

**Formal analysis:** Dwita Rian Desandri, Averina Octaxena Aslani, Faqrizal Ria Qhabibi.

**Methodology:** Dwita Rian Desandri, Averina Octaxena Aslani, Faqrizal Ria Qhabibi, Dimitra Nikoletou.

**Supervision:** Dimitra Nikoletou.

**Validation:** Averina Octaxena Aslani.

**Visualization:** Faqrizal Ria Qhabibi.

**Writing – original draft:** Dwita Rian Desandri, Averina Octaxena Aslani, Faqrizal Ria Qhabibi.

**Writing – review & editing:** Dwita Rian Desandri, Averina Octaxena Aslani, Faqrizal Ria Qhabibi, Dimitra Nikoletou.

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
