## [Decision Letter · Decision Letter 0]

16 Mar 2026

PONE-D-26-05846Exercise-based cardiac rehabilitation and atherosclerotic plaque regression in ASCVD: is exercise really a game-changer?  A scoping review of controlled trialsPLOS One

Dear Dr.  Desandri,

Thank you for submitting your manuscript to PLOS ONE. After careful consideration, we feel that it has merit but does not fully meet PLOS ONE’s publication criteria as it currently stands. Therefore, we invite you to submit a revised version of the manuscript that addresses the points raised during the review process. Please submit your revised manuscript by Apr 30 2026 11:59PM. If you will need more time than this to complete your revisions, please reply to this message or contact the journal office at plosone@plos.org. . Please include the following items when submitting your revised manuscript:

If applicable, we recommend that you deposit your laboratory protocols in protocols.io to enhance the reproducibility of your results. Protocols.io assigns your protocol its own identifier (DOI) so that it can be cited independently in the future. For instructions see: https://journals.plos.org/plosone/s/submission-guidelines#loc-laboratory-protocols. Additionally, PLOS ONE offers an option for publishing peer-reviewed Lab Protocol articles, which describe protocols hosted on protocols.io. Read more information on sharing protocols at . Additionally, PLOS ONE offers an option for publishing peer-reviewed Lab Protocol articles, which describe protocols hosted on protocols.io. Read more information on sharing protocols at https://plos.org/protocols?utm_medium=editorial-email&utm_source=authorletters&utm_campaign=protocols..

We look forward to receiving your revised manuscript.

Kind regards,

Hean Teik Ong, FRCP, FACC

Academic Editor

PLOS One

Journal Requirements:

2. Please include captions for your Supporting Information files at the end of your manuscript, and update any in-text citations to match accordingly. Please see our Supporting Information guidelines for more information: http://journals.plos.org/plosone/s/supporting-information..

Additional Editor Comments:

Please make minor revision to address the comments of the reviewers.

Reviewer's Responses to Questions

**Comments to the Author**

1. Is the manuscript technically sound, and do the data support the conclusions?

Reviewer #1: Yes

Reviewer #2: Yes

2. Has the statistical analysis been performed appropriately and rigorously? 

Reviewer #1: Yes

Reviewer #2: Yes

3. Have the authors made all data underlying the findings in their manuscript fully available?

Reviewer #1: Yes

Reviewer #2: Yes

4. Is the manuscript presented in an intelligible fashion and written in standard English?

Reviewer #1: Yes

Reviewer #2: Yes

5. Review Comments to the Author

Reviewer #1: Full title: Is ( capital I) exercise really a game-changer?

Strong points of this paper:

1. Stringent criteria to select researches to answer the relevant question. Out of 425 researches only 5 were selected for analysis.

2. The mechanism of regression of atherosclerotic plaque from CR is discussed too —- reduction in inflammatory markers.

3. Synergetic effects of CR and LLTs is emphasised in this paper.

The main limitations of this paper are :

1. Unable to distinguish the role of lipid-lowering therapies in the regression of atherosclerotic plaque (As pointed out by the authors )

2. No hard endpoints and cardiovascular outcomes —- due to short duration follow up

3. Different CR Regimes, hence unable to identify the effective CR Intervention

4. The 5 studies analysed in this paper are either open label RCT (four) or non RCT (one)—- many confounding factors.

Nevertheless this paper studies a very important aspect of role of CR which is usually underutilised for secondary cardiovascular diseases prevention. I think it should be used as a hypothesis generating evidence for researchers to do more researches on this issue using RCT—- larger numbers, long term follow up and perhaps hard endpoints study (cardiovascular disease and death) on top of surrogate endpoints ( imaging studies )

Reviewer #2: his scoping review highlights the promising role of exercise-based cardiac rehabilitation (CR) in promoting atherosclerotic plaque regression. By utilizing modalities like HIIT and resistance training alongside standard medical therapy, CR demonstrated significant reductions in plaque volume and improved inflammatory markers across five trials.

However, the review’s conclusions are weakened by critical missing factor of selected studies: the lack of data on statin adherence. Since lipid-lowering therapies are the primary drivers of plaque regression, failing to control for medication compliance makes it difficult to isolate exercise as the definitive "game-changer." Consequently, while CR is an indispensable tool for secondary prevention, large-scale randomized controlled trials are essential to determine if exercise independently drives regression or merely supports pharmacological intervention.

6. PLOS authors have the option to publish the peer review history of their article (what does this mean?). If published, this will include your full peer review and any attached files.). If published, this will include your full peer review and any attached files.

.

Reviewer #1: No

Reviewer #2: No

---

## [Author Response · Author response to Decision Letter 1]

26 Mar 2026

Responses to Reviewer

Reviewer 1:

Full title: Is (capital I) exercise really a game-changer?

Strong points of this paper:

1. Stringent criteria to select researches to answer the relevant question. Out of 425 researches only 5 were selected for analysis.

2. The mechanism of regression of atherosclerotic plaque from CR is discussed too —reduction in inflammatory markers.

3. Synergetic effects of CR and LLTs is emphasised in this paper.

The main limitations of this paper are :

1. Unable to distinguish the role of lipid-lowering therapies in the regression of atherosclerotic plaque (As pointed out by the authors)

2. No hard endpoints and cardiovascular outcomes — due to short duration follow up

3. Different CR Regimes, hence unable to identify the effective CR Intervention

4. The 5 studies analysed in this paper are either open label RCT (four) or non RCT (one)—many confounding factors.

Nevertheless this paper studies a very important aspect of role of CR which is usually underutilised for secondary cardiovascular diseases prevention. I think it should be used as a hypothesis generating evidence for researchers to do more researches on this issue using RCT— larger numbers, long term follow up and perhaps hard endpoints study (cardiovascular disease and death) on top of surrogate endpoints (imaging studies)

Response:

The authors wish to express their sincere gratitude to the reviewer for their constructive insights and comprehensive evaluation of both the strengths and the primary limitations of this manuscript. We have carefully addressed each point raised to enhance the clarity and rigor of the work.

Response to Technical and Formatting Corrections:

The authors have corrected the typographical error in the title as identified. Specifically, the notation has been amended to the formal capitalized (I).

Response to Study Limitations:

Regarding the substantive limitations noted by the reviewer, the authors have conducted a thorough re-evaluation of the data and the existing literature. We fully acknowledge these constraints and have significantly expanded the Study Limitations section to provide a more critical and transparent discussion of the findings. These revisions can be found on pages 14–15, lines 231–240, “There are several limitations in this review. Firstly, the current analysis was unable to independently distinguish the specific role of concomitant LLTs in the observed regression of atherosclerotic plaque, a factor that remains a potential confounder in evaluating the primary intervention. Lack of information on drug adherence prevents us from determining whether plaque regression is directly attributable to CR. Second, the relatively short duration of follow-up across the included studies precluded the assessment of clinical endpoints and long-term cardiovascular outcomes. Third, significant heterogeneity in the CR regimens utilized across the literature makes it challenging to identify the most effective specific CR intervention. Lastly, although grey literature was considered, no relevant records were identified.”

Regarding the identified study limitations, the authors have further enhanced the manuscript’s academic contribution by incorporating a dedicated discussion on Future Research Directions. This newly added paragraph addresses the current gaps in the literature and outlines a strategic framework for subsequent investigations. These revisions are detailed on page 15, lines 241–250, “Based on the limitations identified in this study, future research should prioritize several key areas to bridge the current gaps in cardiovascular rehabilitation and plaque management, including (1) dissecting therapeutic synergy to accurately isolate the regression of atherosclerotic plaque. Future trials must be designed to distinguish the independent and effects of LLTs versus lifestyle-based CR intervention. (2) Long-term outcome data, there is a critical need for studies with extended follow-up duration in the aim to obtained hard clinical endpoints, including MACE, cardiovascular mortality, and all-cause hospitalization. (3) Future evidence synthesis would benefit from a higher volume of double-blind and multicenter RCTs, providing a clearer picture of the intervention’s true efficacy. (4) Leveraging high-resolution intravascular imaging in future cohorts could provide deeper insights into plaque morphology and stability beyond simple volume regression.”

Reviewer 2:

This scoping review highlights the promising role of exercise-based cardiac rehabilitation (CR) in promoting atherosclerotic plaque regression. By utilizing modalities like HIIT and resistance training alongside standard medical therapy, CR demonstrated significant reductions in plaque volume and improved inflammatory markers across five trials.

However, the review’s conclusions are weakened by critical missing factor of selected studies: the lack of data on statin adherence. Since lipid-lowering therapies are the primary drivers of plaque regression, failing to control for medication compliance makes it difficult to isolate exercise as the definitive "game-changer." Consequently, while CR is an indispensable tool for secondary prevention, large-scale randomized controlled trials are essential to determine if exercise independently drives regression or merely supports pharmacological intervention.

Response:

The authors wish to express their sincere appreciation for the reviewer’s constructive and insightful comments regarding these critical points. Related to the issues that mentioned above, the authors have been studied and investigate thoroughly. Therefore, in facing the issues the authors have been modified the limitation contents on pages 14–15, lines 233 – 235 related to the lack of data on statin adherence. Furthermore, to preserve the thematic context of the 'game-changer' hypothesis while responding to the reviewer’s concerns, we have incorporated a dedicated Future Research Directions section (page 15, lines 241–250). This addition serves to outline a strategic framework for subsequent investigations to systematically accommodate and mitigate the issues constraints identified in the present study.

---

## [Decision Letter · Decision Letter 1]

5 Apr 2026

Exercise-based cardiac rehabilitation and atherosclerotic plaque regression in ASCVD: Is exercise really a game-changer?  A scoping review of controlled trials

PONE-D-26-05846R1

Dear Dr. Dwita Rian Desandri,

Congratulations, your article has been accepted by the academic editor and reviewers.  You now have to meet requirements from the administrative editors, if any.

We’re pleased to inform you that your manuscript has been judged scientifically suitable for publication and will be formally accepted for publication once it meets all outstanding technical requirements.

An invoice will be generated when your article is formally accepted. Please note, if your institution has a publishing partnership with PLOS and your article meets the relevant criteria, all or part of your publication costs will be covered. Please make sure your user information is up-to-date by logging into Editorial Manager at Editorial Manager® and clicking the ‘Update My Information' link at the top of the page. For questions related to billing, please contact  and clicking the ‘Update My Information' link at the top of the page. For questions related to billing, please contact billing support..

Kind regards,

Hean Teik Ong, FRCP, FACC

Academic Editor

PLOS One

Additional Editor Comments (optional):

Congratulations, your article has been accepted by the academic editor and reviewers. You now have to meet requirements from the administrative editors, if any.

Reviewers' comments:

Reviewer's Responses to Questions

**Comments to the Author**

1. If the authors have adequately addressed your comments raised in a previous round of review and you feel that this manuscript is now acceptable for publication, you may indicate that here to bypass the “Comments to the Author” section, enter your conflict of interest statement in the “Confidential to Editor” section, and submit your "Accept" recommendation.

Reviewer #1: All comments have been addressed

Reviewer #2: (No Response)

2. Is the manuscript technically sound, and do the data support the conclusions?

Reviewer #1: Yes

Reviewer #2: (No Response)

3. Has the statistical analysis been performed appropriately and rigorously? 

Reviewer #1: Yes

Reviewer #2: (No Response)

4. Have the authors made all data underlying the findings in their manuscript fully available?

Reviewer #1: Yes

Reviewer #2: (No Response)

5. Is the manuscript presented in an intelligible fashion and written in standard English?

Reviewer #1: Yes

Reviewer #2: (No Response)

6. Review Comments to the Author

Reviewer #1: No further comment All my concerns are addressed.

The paper raised a good point to remind the clinicians the importance of incorporating exercise in the process of helping patients to prevent ASCVD.

Please publish accordingly

Reviewer #2: (No Response)

7. PLOS authors have the option to publish the peer review history of their article (what does this mean?). If published, this will include your full peer review and any attached files.). If published, this will include your full peer review and any attached files.

.

Reviewer #1: No

Reviewer #2: No

---

## [Editor Report · Acceptance letter]

PONE-D-26-05846R1

PLOS One

Dear Dr. Desandri,

I'm pleased to inform you that your manuscript has been deemed suitable for publication in PLOS One. Congratulations! Your manuscript is now being handed over to our production team.

Kind regards,

on behalf of

Dr. Hean Teik Ong

Academic Editor

PLOS One